# Abandonment and Recultivation of Agricultural Lands in Slovakia—Patterns and Determinants from the Past to the Future

**Robert Pazúr** [1,2,*] **, Juraj Lieskovský** [3] **, Matthias Bürgi** [1,4] **, Daniel Müller** [5,6,7] **,**
**Tibor Lieskovský** [8] **, Zhen Zhang** [9] **and Alexander V. Prishchepov** [10,11]

1. Swiss Federal Institute for Forest, Snow and Landscape Research WSL, Zürcherstrasse 111, CH-8903 Birmensdorf, Switzerland; matthias.buergi@wsl.ch
2. Institute of Geography, Slovak Academy of Sciences, Stefanikova 49, 814 73 Bratislava, Slovakia
3. Institute of Landscape Ecology, Slovak Academy of Sciences, Akademická 2, 949 01 Nitra, Slovakia; Juraj.Lieskovsky@savba.sk
4. Institute of Geography, University of Bern, CH-3012 Bern, Switzerland
5. Leibniz Institute of Agricultural Development in Transition Economies (IAMO), Theodor-Lieser-Strasse 2, 06120 Halle (Saale), Germany; d.mueller@hu-berlin.de
6. Geography Department, Humboldt Universität zu Berlin, Unter den Linden 6, 10099 Berlin, Germany
7. Integrative Research Institute on Transformations of Human-Environment Systems (IRI THESys), Humboldt Universität zu Berlin, Unter den Linden 6, 10099 Berlin, Germany
8. Department of Theoretical Geodesy, Faculty of Civil Engineering, Slovak University of Technology in Bratislava, Radlinskeho 11, 810 05 Bratislava, Slovakia; tibor.lieskovsky@stuba.sk
9. Department of Geographical Sciences, University of Maryland, College Park, MD 20742, USA; yuisheng@gmail.com
10. Department of Geosciences and Natural Resource Management (IGN), University of Copenhagen, Øster Voldgade 10, DK-1350 København K, Denmark; alpr@ign.ku.dk
11. Institute of Steppe of the Ural Branch of the Russian Academy of Sciences, Pionerskaya str. 11, 460000 Orenburg, Russia

\* Correspondence: robert.pazur@wsl.ch

**Abstract:** Central and Eastern Europe has experienced fundamental land use changes since the collapse of socialism around 1990. We analyzed the patterns and determinants of agricultural land abandonment and recultivation in Slovakia during the transition from a state-controlled economy to an open-market economy (1986 to 2000) and the subsequent accession to the European Union (2000 to 2010). We quantified agricultural land-use change based on available maps derived from 30-m multi-seasonal Landsat imagery and analyzed the socioeconomic and biophysical determinants of the observed agricultural land-use changes using boosted regression trees. We used a scenario-based approach to assess future agricultural land abandonment and recultivation until 2060. The maps of agricultural land use analysis reveal that cropland abandonment was the dominant land use process on 11% of agricultural land from 1986 to 2000, and on 6% of the agricultural land from 2000 to 2010. Recultivation occurred on approximately 2% of agricultural land in both periods. Although most abandoned land was located in the plains, the rate of abandonment was twice as high in the mountainous landscapes. The likelihood of abandonment increased with increased distance from the national capital (Bratislava), decreased with an increase of annual mean temperatures and was higher in proximity to forest edges and on steeper slopes. Recultivation was largely determined by the opposite effects. The scenario for 2060 suggests that future agricultural land abandonment and recultivation may largely be determined by climate and terrain conditions and, to a lesser extent, by proximity to economic centers. Our study underscores the value of synergetic use of satellite data and land-use modeling to provide the input for land planning, and to anticipate the potential effects of changing environmental and policy conditions.

**Keywords:** land cover; land use; land use intensity; remote sensing; machine learning; Landsat; land change drivers; Eastern Europe; socialism; scenario; climate

## 1. Introduction

Agricultural land use provides essential food and fiber resources, bioenergy products and non-material benefits. However, changes in agricultural land use have been the main cause of global environmental change, land degradation, biodiversity losses and the reduction of ecosystem services [1]. In the tropics, agricultural expansion is responsible, besides other consequences, for massive emissions of greenhouse gases and biodiversity decline [2]. In temperate regions, however, widespread agricultural land abandonment (i.e., discontinuation or a significant decline of active use of the agricultural land, which results in grassland, shrubs and trees encroachment on former croplands) has often fostered the restoration of ecosystem services, including the habitat for umbrella species [3–5]. Better knowledge about the factors that determine the spatial-temporal patterns of land use change may help to increase awareness and allow for better targeting of societal and policy responses to land use change.

Central and Eastern Europe (CEE) has been one of the global hotspots of agricultural land abandonment in recent decades [6–8]. The political overhaul in 1989–1990 resulted in the demise of state-governed socialist systems, and the subsequent transformation of the agricultural sector from state-control toward market economies. All countries in CEE reorganized their agricultural sectors and implemented land reforms that shifted land ownership from the collective back to individual users, resulting in the subsequent abandonment of agricultural lands [9–12]. The rates and patterns of agricultural land abandonment differed substantially across post-socialist countries because of the differences in the speed and depth of the implementation of the reforms following the demise of Soviet socialism [13–15]. Abandonment tended to be higher in socio-economically and agro-environmentally marginal areas, for instance, in areas located further away from settlements and potential markets, in mountainous regions, in areas with lower soil fertility and in less densely populated regions [16–22].

The second wave of agricultural land use change was triggered by the accession of some CEE countries to the European Union (EU) [23,24]. The contribution of EU funds for agricultural and rural development favored recultivation of some abandoned agricultural lands [25], especially in less-favored areas (LFAs) that received considerable support under the common agricultural policy (CAP) [26–28]. However, existing empirical evidence on the land use trajectories across CEE mainly rests on analyzes from studies conducted on small areas, and very few account for the effect of EU policies on subsequent land use changes. Moreover, while evidence of the effect of different policies may be hidden in aggregated statistical data, it is likely that such effects can be assessed using spatially-explicit data sources (e.g., produced with satellite observations).

Abandonment after the collapse of socialism and recultivation during EU accession are two major land use change processes that have shaped agricultural landscapes in Slovakia [21,23,29]. Approximately 18% of parcels designated as agricultural land in 1990 (4300 km$^2$) were partially or completely overgrown by shrubs and young trees by 2010 [24,30]. Abandonment primarily affected small-scale traditional agricultural landscapes [31–33] with a high risk of adverse impacts on biodiversity and on cultural and historical heritage [34–36]. A second major land-change process is the conversion of grasslands into croplands in the EU pre-accession and post-accession periods due to shifting production toward more profitable crops [24] and EU supporting schemas (e.g., SAPARD, CAP) [37].

Previous attempts to map the country-wide spatial distribution of agricultural land use change in Slovakia relied on the Coordination of Information on the Environment (CORINE) land cover products for the years 1990, 2000, 2006 and 2012 [21,38,39]. The CORINE dataset maps European land use classes with a nomenclature that is purposely limited to relatively coarse spatial resolution (a minimum

mapping unit of 25 ha, and a minimum resolution of 5 ha for temporal change mapping). Such coarse spatial and temporal levels lead to the omission of small-scale changes (i.e., changes of less than 5 ha). In Slovakia, however, many areas, particularly in mountainous regions, are characterized by small agricultural patches embedded in forests that are often prone to abandonment [40,41]. Such patterns are elusive because no wall-to-wall mapping using satellite imagery at a high spatial resolution has been conducted to date.

Examples from Albania [42], Western Ukraine [43], European Russia [20], Kazakhstan [44–46] and CEE countries [13,47–49] proved the suitability of multispectral 30-meter historical records of Landsat imagery to accurately map agricultural land use change. Multi-seasonal Landsat imagery was particularly useful to separate managed and non-managed agricultural fields [50]. One such agricultural land use change map produced with a pixel-based image compositing approach was developed from 1985 to 2010 for the Carpathian region, including the entire area of Slovakia [27]. This product is ideally suited to study the patterns and determinants of agricultural land abandonment and recultivation in Slovakia because of its high accuracy, fine spatial details and long temporal coverage.

A scenario-based approach can be useful to analyze alternative future trajectories of land use change [51]. Scenarios may represent qualitative storylines that capture the changes in salient determinants into the future. For example, the future development of agriculture can be determined by political, economic, technological and environmental changes. Quantification of these drivers then permits land use change to be forecasted [52].

In the present study, we aimed to understand the spatial patterns and determinants of agricultural land use change in Slovakia from 1985 to 2010, as well as to make projections about agricultural development until 2060. We used 30-meter Landsat-based agricultural land use change maps from 1985 to 2010 build on previous work [27] and a set of spatially-explicit socioeconomic and biophysical variables. We developed a scenario of plausible agricultural land use change processes for 2060, assuming similar trends in the agricultural land change to those observed in the past and a future climate-change projection. Using the scenario output, we sought to better understand the spatial consequences of agricultural land use change and highlight areas that will likely experience an agricultural land use change in future decades. Our specific objectives were to:

(1)  describe the patterns of agricultural land use change in Slovakia using Landsat-based maps for 1985–2000 (transition period) and 2000–2010 (EU-accession);
(2)  quantify the influence of determinants on abandonment and recultivation of agricultural land for both periods;
(3)  assess agricultural land use change until 2060 based on a land change scenario.

## 2. Data and Methods

### 2.1. Study Area

Slovakia is located in Central Europe (Figure 1) and comprises an area of 49,036 km$^2$ with a population of 5.5 million people (LAU2); 54.3% of Slovakia's population lives in cities [53]. Elevation varies from 85 meters in the south-east up to 2655 meters in the north. The landscape can be divided into two geomorphological parts, which also reflect the specialization of agriculture: the plain (lowlands and basins in south-western, southern and eastern Slovakia) and the mountainous region (central and northern Slovakia) [54]. The plain comprises the most fertile soils: haplic luvisol, fluvisols and chernozems (IUSS Working Group WRB, 2007). Both economically developed areas and a concentration of agricultural production areas are found in the Danubian Lowland, located in the south-west of Slovakia. Overall, the climate is moderately cool and humid. In the plains, the annual mean temperature for 1985–2010 ranged from 3.7 °C to 11.7 °C (mean of 9.3 °C) and the annual precipitation from 528 mm to 1162 mm (mean 636 mm), respectively. In mountainous regions, the annual mean temperature and amount of rainfall ranged from −4.8°C to 10.9° (mean of 6.8 °C) and from 534 mm to 1867 mm (mean of 802 mm) for 1979–2015, respectively.

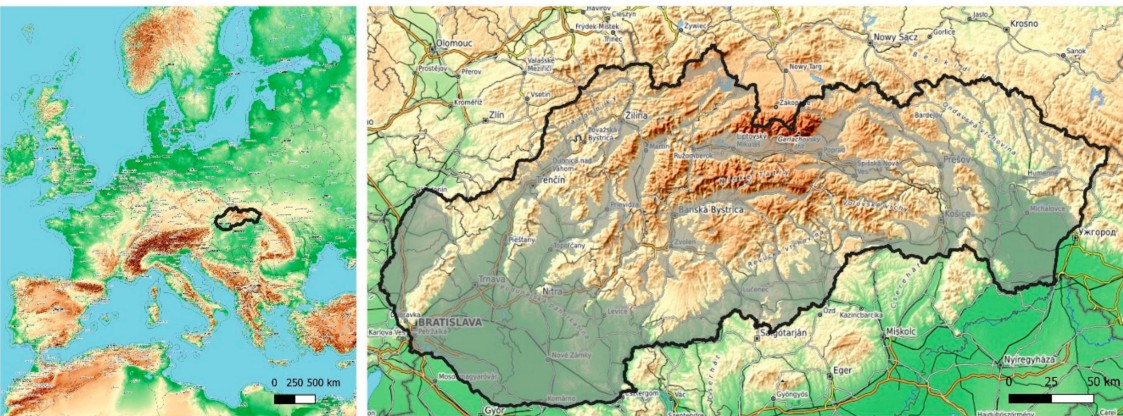

**Figure 1.** The study area of Slovakia. The grey color covers the area considered as being lowlands and basins. Source: OpenTopoMap (https://opentopomap.org); [54].

A massive land transformation of the 20th century occurred in Slovakia during the Socialist period (1946–1989). Agricultural fields were nationalized and enlarged to make them suitable for large-scale collective farming [12,55,56]. Between 1960 and 1989, the number of employees in agriculture decreased by 31% to 351,000 workers, of which more than three quarters were employed in collective farms [57]. During the late socialism period (1980–1990) former agricultural workers found jobs in the growing light and heavy industry in the Slovak Socialist Republic [58]. At the same time, only 0.4% of agricultural land experienced the conversion from grassland to cropland, a sign of the intensification of agricultural production [58]. Conversion to a market-oriented economy in the 1990s brought a general decline in the agricultural sector. For example, the number of pigs decreased by 77% from 1980 to 2010, and the amount of sown feed crops fell sharply, leading to agricultural land abandonment (Figure 2). The number of people employed in agriculture decreased from 351,000 in 1989 to merely 51,500 in 2010 [57]. The removal of trade barriers and subsidies in recent decades influenced different sectors of agricultural production. Vine production, for example, was severely hit, as more than 1/3 of all vineyards were abandoned or converted into cropland [59]. Following the land restitution phase, agricultural parcels were often given back to people who do not live on or from the land, that is, those who had migrated to cities, or did not have any specific interest in farming [60]. With the accession to the EU in 2004, the demand for energy crops fostered an increase in cultivated areas with oil plants, partially substituting for other crops and expanding on formerly abandoned lands. In summary, structural changes in agricultural production and institutional changes with respect to land use have resulted in multiple trajectories of agricultural land use change over the past three decades.

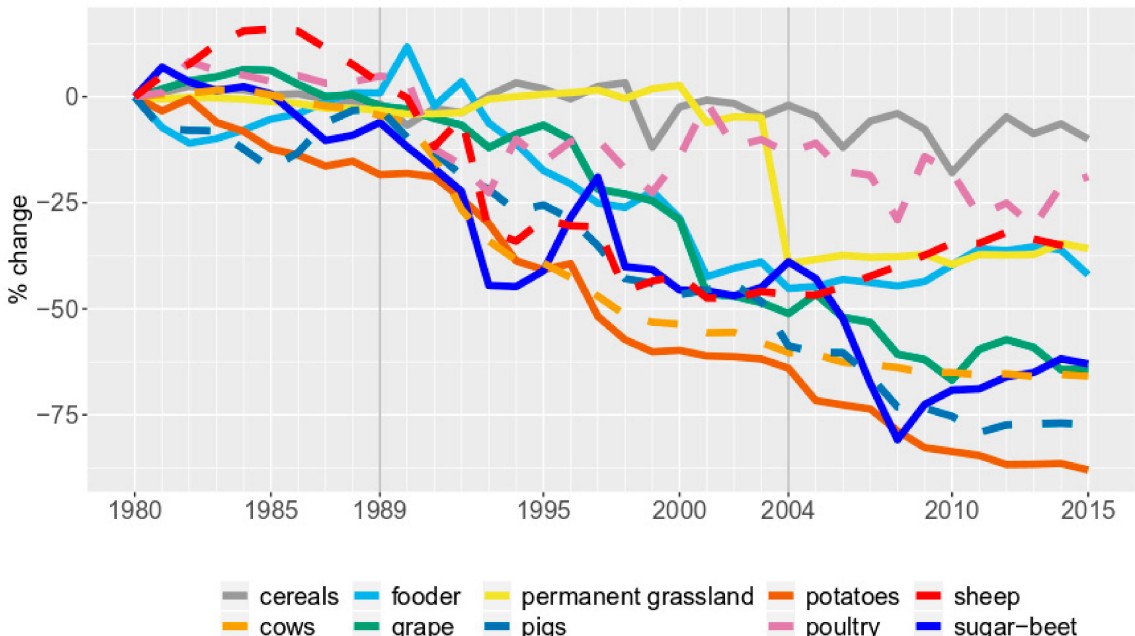

**Figure 2.** Relative change in crop yields (solid lines) and livestock numbers (dashed lines) from 1980 to 2015. The vertical grey lines indicate the years of the fall of socialism and accession to the European Union. Source: Statistical Office of the Slovak Republic.

## 2.2. Mapping Land Use Change

We utilized 30-m land cover maps for 1985, 2000 and 2010 that were generated in the previous study, when multi-seasonal Landsat imagery were classified with a random forest classifier for the Carpathian Ecoregion, including Slovakia [27]. Produced land cover maps distinguished cropland, built-up, forest, grassland with the purpose of analyzing trajectories of agricultural land use change.

The reported overall accuracy of the agricultural land use change map was 90% with user's and producer's accuracies ranged from 34% to 96% and from 11% to 94% according to the different agricultural land use change process [27].

Here, we specifically looked at two agricultural land use change processes, namely, agricultural land abandonment and recultivation of abandoned agricultural lands that occurred in two time-steps: the period of postsocialism (1985–2000; period #1) and the period of accession to the European Union (2000–2010; period #2). The agricultural land abandonment class was assigned when actively managed cropland was transformed into grassland or if both managed croplands and grasslands underwent conversion to forest. The conversion of croplands to grasslands was considered as the abandonment of agricultural activities, however, we should not exclude the fact that some areas represented a reduced intensification (e.g., longer fallow, reduced grazing). In contrast, all grassland areas converted to cropland and all forest converted to either grassland or cropland were assigned to the recultivation class.

## 2.3. Determinants of Agricultural Land Use Change

We selected variables that may potentially impact farmers' decisions to either abandon or recultivate a plot of land [61–63]. Selected determinants (Table 1) included variables collected from aggregated administrative-level statistics and spatially-explicit pixel-level data.

We assumed that the biophysical conditions determined the overall suitability of land for agricultural production. A set of biophysical properties (Table 1) was derived from the digital elevation model (DEM), soil map and climate data. All spatially-explicit datasets were resampled to match the 30 m resolution of Landsat-based land use change maps. We used the aggregated 30 m dataset of DEM originally interpolated to the 15-m grid from civil contour maps (1:10,000) [64]. Sun radiation was

calculated from the global radiation model [65]. The topographic position index (TPI) [66] classifies landscapes into landform categories (e.g., steep, narrow canyon, gentle valley, plain, open slope, mesa). We derived TPI index values from the direct Moore (8 pixel connectivity) neighborhood pixels (30 × 30 m pixel size), which we found to be optimal for the classification of most surface distortions. To assess soil quality, we used the topographic wetness index (TWI) as a proxy for soil moisture and derived it from the slope and specific catchment area [67]. To distinguish between agricultural lands in mountainous and flatland areas, we used a dummy variable (binary value of inclusion/exclusion of a particular pixel in mountainous areas). Soil properties were represented by soil fertility using the relative scoring according to [68], which derives soil fertility as a function of climate, soil type, particle size distribution, soil depth, slope steepness and aspect. In our case, we considered only particle size distribution and soil type because the effects of slope steepness and climate on the abandonment and recultivation of agricultural land were analyzed separately and the soil depth parameter was not available for the entire study region. As the input data for soil parameters on agricultural soils, we used the soil map officially produced by the Slovakian National Agriculture and Food Center, which is based on soil samples extrapolated to geographically homogenous units [68]. For forest soils, we used the data from forest soil maps provided by the National Forest Center of Slovakia. Climatic conditions were represented by the mean temperature and rainfall during the season (April–September). We used the CHELSA high-resolution climate data set [69] for 1979–2015 and downscaled the initial resolution 30 arcsec (~1 km) data to 30 m by using the nearest neighborhood transformation and the DTM model.

　　　　We also took the special status of protected natural areas into consideration in our models and identified areas with a history of long-term nature protection encompassing higher protection grades (from 2nd—less protected to 5th—strict protection) according to the Natura 2000 database [70] and areas protected under the Ramsar Convention [64]. For analyses of past changes (1985–2010), we used the nature protected areas only as an independent dummy variable. For the future land-change scenario (2060), we assumed such areas would remain stable and did not allow any land change to be allocated to them.

　　　　We assumed that physical accessibility as an important factor that affects land use [19,71]. To assess the influence of travel costs, we calculated several distance-based measures. For settlement accessibility-related determinants, we used a cost-distance approach that incorporates the effect of topographic and land use factors that have a barrier effect (buildings, rivers), or could hinder (meadows, fields, forests) or facilitate (paved roads, paths, bridges) the movement of material and information. Topography and the barrier effects of the different land use classes were expressed as friction coefficients. The friction coefficients of different land use types were derived from empirical data and published sources [72,73].

　　　　We considered the cost-distance to settlements as a walking distance, and we incorporated the effect of topography (uphill or downhill) on walking speed. To calculate the distance from the capital city by car, we used the speed coefficients adjusted for road network speed limits.

　　　　We also analyzed population characteristics that are potential factors in agricultural land use change decisions. This data was collected at the commune level by the Statistical Office of the Slovak Republic (2015) (Table 1). We included population census data from the years 1981, 2000, 2011 and calculated relative population change, which allowed us to compare the values across communes and evaluate their principals of marginality in terms of economy (changes in population density, aging of the population, economic activity, migration rates, construction of new housing facilities) and agricultural production (the proportion of people working in agriculture).

## 2.4. Boosted Regression Trees

　　　　Spatially-explicit modeling approaches are commonly used to assess the spatial determinants of agricultural land use change [74]. While ordinary least squares linear regression and logistic regression models are often used [75–77], improvements in computational power and algorithms have resulted in non-parametric machine-learning techniques, such as boosted regression trees, becoming increasingly

popular. Boosted regression trees are useful to capture nonlinearities in the potential influence of the determinants of agricultural land abandonment [19,78,79].

The analysis of potential causes and consequences of agricultural land use change are often complemented with the use of cross-section regression models [75]. Fitting parametric regression models requires a targeted sampling design [80] that helps to establish a linear relationship between predictors and the outcome variable, to remove mutual collinearity of variables, or, in the case of spatial data, to address the potential presence of spatial autocorrelation. Alternatively, algorithmic models, such as boosted regression trees (BRTs), may be developed without any a priori knowledge about the relationship between the target and the predictor variable [81]. BRTs are a machine learning algorithm that relies on the boosting technique to combine many relatively simple regression trees. BRTs are ideally suited to analyzing determinants of land change because they can model non-linear relationships, efficiently select relevant predictors, they are insensitive to outliers, automatically account for interactions between predictors and combine high predictive accuracy with good interpretability of input-output relationships ([82–85]. As with other regression models, BRTs make it possible to analyze the occurrence of different events in a probabilistic framework, quantify the relative contribution of predictors to the observed phenomenon and establish the effect of individual predictors on the outcome (in our case, abandonment or recultivation).

In the present study, we used BRTs (using the "dismo" R- package [86]) to model agricultural land use change with spatial determinants that are hypothesized to shape the location of the observed changes. The models then establish the statistical relationship between the agricultural land use change as the target outcome variable and specific explanatory predictors (spatial determinants), the strength of the relationship and the non-linear dependency of agricultural land use change on variation in the value of the particular factor (e.g., how an increasing slope influences the likelihood of abandonment).

Although the BRTs are able to cope with the mutual correlation of the explanatory predictors used to fit the model, a close relationship of these predictors may influence their explanatory power (e.g., low explanatory power of the precipitation rates when considered together with the average temperature). From the analysis, we, therefore, excluded those explanatory predictors that in the pairwise comparison received the Pearson correlation value higher than 0.7. From each pair, we further considered only one variable that did not correlate with multiple other predictors and was more meaningful for explaining the patterns of change on the agricultural land (Figure S1, Table 1).

*2.5. Parameterization of Boosted Regression Trees*

Evaluating nationwide land use change data at a 30-m spatial resolution with BRTs requires a sampling strategy to reduce the total number of observations and to avoid spatial autocorrelation. Thus, we applied stratified balanced sampling by randomly selecting 5000 points with the presence of agricultural land change, and an equal number with the absence of agricultural land change, representing two strata, respectively. The absence of agricultural land use change strata included both the total extent of agricultural areas (representing the absence of abandonment) and the extent of grassland and forests (representing areas that can be recultivated).In addition to the sampling procedure, BRT outputs largely depend on the settings of tree complexity levels (number of branches or splits)levels and learning rates. Suitable values for both parameters can be found by comparing BRT model results with a different set of these parameters. We assessed model sensitivity by testing all combinations of tree complexity levels from 1 to 9 (with a stepwise increase of 1) with learning rates ranging between 0.1 to 0.001 (with a stepwise increase of 0.005). The resulting models were selected based on accuracy, which we calculated with a 10-fold cross-validated correlation coefficient [58,81]. To evaluate overall model accuracy, we assigned 80% of the samples to the training dataset and 20% to the testing dataset. Finally, we calculated the area under the receiver operating curve (AUC) [86,87] to assess the proportion of errors between mapped and predicted areas.

*2.6. Modeling Future Land Use Change*

To understand potential future agricultural land use change, we constructed a trend scenario for the year 2060. In the scenario, we assumed a similar amount of abandonment and recultivation of agricultural land as between 2000 and 2010. The target period of our scenario model was up to 2060, as such a length could be sufficient to observe the predicted climate-change effects. To determine the extent and location of agricultural abandonment and recultivation in the future, we constructed an allocation model. This model allocates abandonment and recultivation at similar area size as observed in 2000–2010 (period #2). For the allocation, we used the probability maps derived from the BRTs models of agricultural land use change assuming similar strength and influence of spatial determinants as fitted by BRT models for 2000–2010. We iterated the model over ten-year time periods (starting with 2010 until 2060) while at each iteration, taking the land use configuration from the previous time step into consideration and newly derived set of probability maps. At each iteration (10-years time steps) we allocated the areas of abandonment and recultivation (same area size observed in 2000–2010) within the pixels of the highest probability values on the probability map of abandonment and recultivation. Pixels that were considered as abandoned during the allocation process were set as not available for the recultivation model.

The explanatory variables used to derive the probability maps of abandonment and recultivation were considered to be static over time, except for the distances to agricultural and forested land (which changed with respect to the initial map of agricultural land use), and the climate variables. The predicted mean temperature and precipitation during the growing season in 2060 were derived from Community Earth System Model Atmosphere Model version (CESM-CAM5) outputs under the Representative Concentration Pathways (RCP) 8.5 scenario in IPCC AR5 [88], which assume high greenhouse-gas emissions with high population increase and no climate-mitigation policies in place. From the areas available for the future land use change, we excluded protected areas (keeping the land use in protected areas fixed from 2010 on) and allowed forested land to be converted to grassland (a land change which was not observed in the period 2000–2010).

**Table 1.** Analyzed explanatory variables.

| Variables | Abbreviation | Characteristics | Data Source |
|---|---|---|---|
| **Biophysical** | | | |
| elevation * | elevation | DEM | [64] |
| slope | slope | DEM | [64] |
| sun radiation | solar | solar radiation model (kWh) [65] | [64] |
| topographic position index | TPI | [66] | [64] |
| topographic wetness index | TWI | [66] | [64] |
| relief subgroup | lowland | areas that belongs to the flatland or mountain relief | [54] |
| soil fertility | fertility | function of grain size distribution and soil type | [89] |
| protected sites | protection | all natural protected areas except large-area national parks | [64] |
| **Climatic** | | | |
| temperature | temp | monthly means of temperature in 1985–2000 and 2000–2010 | [90] |
| rainfall * | rainfall | monthly means of rainfall in 1985–2000 and 2000–2010 | [90] |
| **Accessibility &Isolation** | | | |
| distance to forested area | forest | cost-distance using DEM | [64], own calculations |
| distance to capital city | capital | cost-distance in minutes using the friction coefficients for different LU classes and DEM | [64], own calculations |
| distance to important cities | regional3 | cost-distance in minutes using the friction coefficients for different LU classes and DEM | [64], own calculations |
| distance to regional centers | regional8 | cost-distance in minutes using the friction coefficients for different LU classes and DEM | [64], own calculations |
| distance to LAU 1 (NUTS 4) centers | LAU | cost-distance in minutes using the friction coefficients for different LU classes and DEM | [64], own calculations |
| distance to communes * | settlement | cost-distance in minutes using the friction coefficients for different LU classes and DEM | [64], own calculations |
| distance to roads | road | cost-distance using DEM | [64], own calculations |
| **Demographic** | | | |
| population density | pop_dens | population/km$^2$ periods (1985–2000, 2000–2010) | [53] |
| age index | age | population over 65 y./population under 15y.* 100 (1985–2000, 2000–2010) | [53] |
| migration | migration | total migration/population (1985–2000, 2000–2010) | [53] |
| economic activity | ec_active | proportion of economic active people (1985–2000, 2000–2010) | [53] |
| agriculture activity | agri_active | proportion of people working in agriculture (1985–2000, 2000–2010) | [53] |
| flats | flats | flats/apartments build in previous decade/hectares (1985–2000, 2000–2010) | [53] |

* this variable was removed prior to the modeling due to high mutual correlation with another variable from the list.

## 3. Results

### 3.1. Past Agricultural Land Use Change

The two agricultural land use change processes considered, namely, abandonment and recultivation affected 4763 km² (8% of the total area of Slovakia and 19% of all cultivated agricultural lands in 1985) from 1985 to 2010, with abandonment being the dominant agricultural land use change process (Figure 3). During the transition period (1985–2000; period #1) 11% of the cultivated agricultural land in 1985 became abandoned by 2000 (Figure 3). During the following EU-related period (2000–2010; period #2) another 6% of cultivated agricultural land in 2000 became abandoned. Agricultural lands became predominantly abandoned in plains with 63% of the country-wide abandoned area during the first period (1985–2000) and 75% during the second period (2000–2010). However, from the total agricultural area, the rate of abandonment was twice as high in the mountainous area (14% from 1985 to 2010) compared to the plains (7% from 1985 to 2010) (Figure 4).

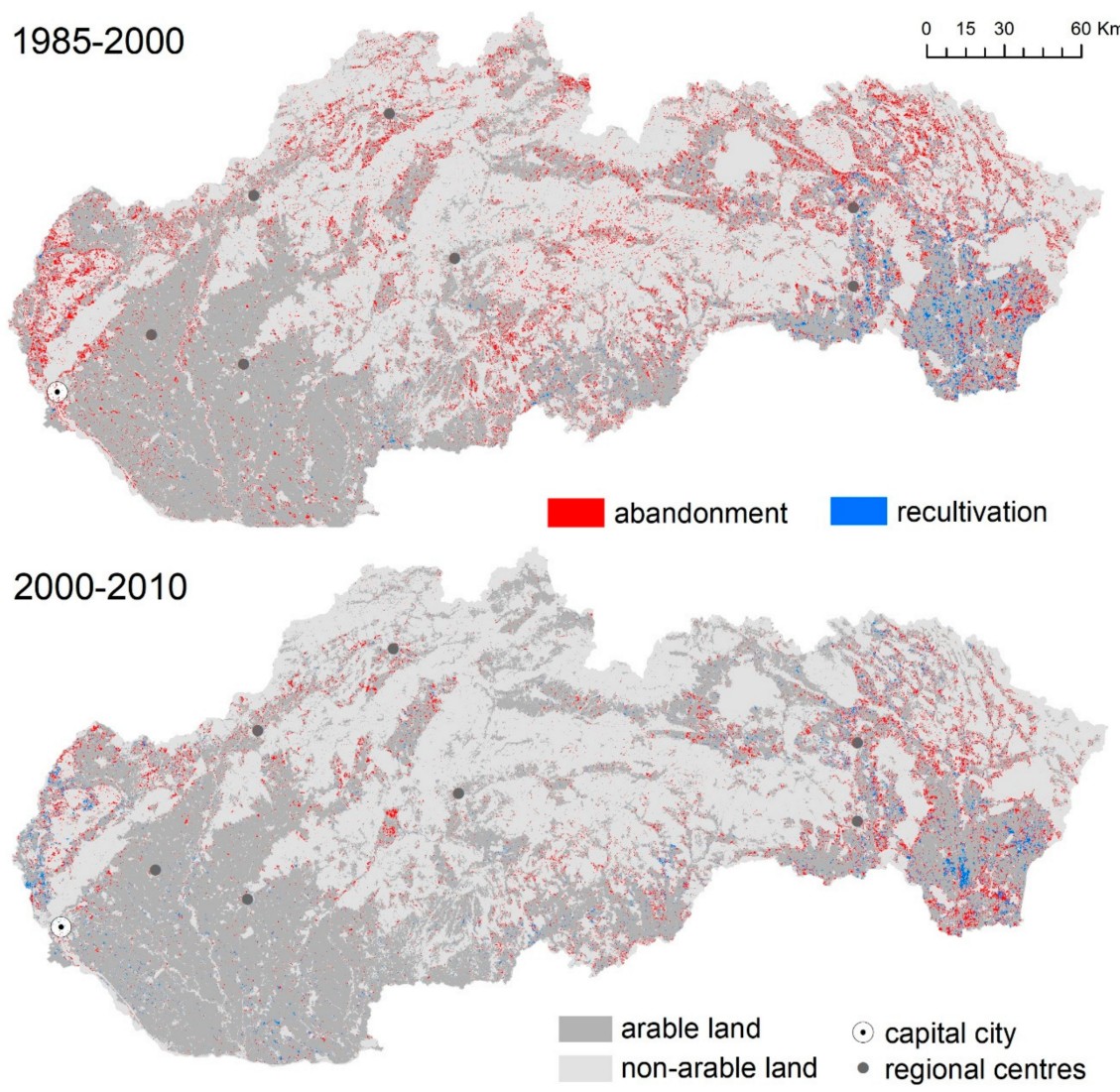

**Figure 3.** Agricultural land use change in Slovakia for 1985–2000 and 2000–2010. Source: [27].

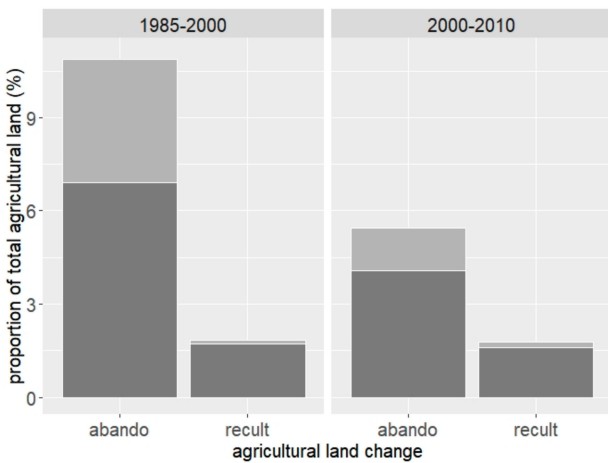

**Figure 4.** The proportion of agricultural land use in flatland (dark grey color) and mountain regions (light grey color) influenced by abandonment (abando) and recultivation (recult).

Spatial patterns of land abandonment were best explained by distance-based and climate-related factors (Figure 5). Abandonment in both periods primarily occurred in areas close to the forest. In western Slovakia, the probability of abandonment decreased with increased distance from Bratislava, whereas in central and eastern Slovakia, the opposite effect was observed regarding the influence of distance from Bratislava (Figures 3 and 5). This west-east gradient dominated especially in period #2, when the distance to capital city (Bratislava) accounted for 14% of the variance in the occurrence of abandonment. Temperature was the most important biophysical variable in explaining abandonment of agricultural land in period #1 and the third most important in period #2. The probability of abandonment also increased with increasing slope, terrain roughness (TPI) and lower solar radiation. The proportion of people working in agriculture explained less than 5% of the variance in abandonment. The accuracy of the abandonment models, quantified by AUC was 0.76 for period #1 and 0.83 for the period #2, respectively.

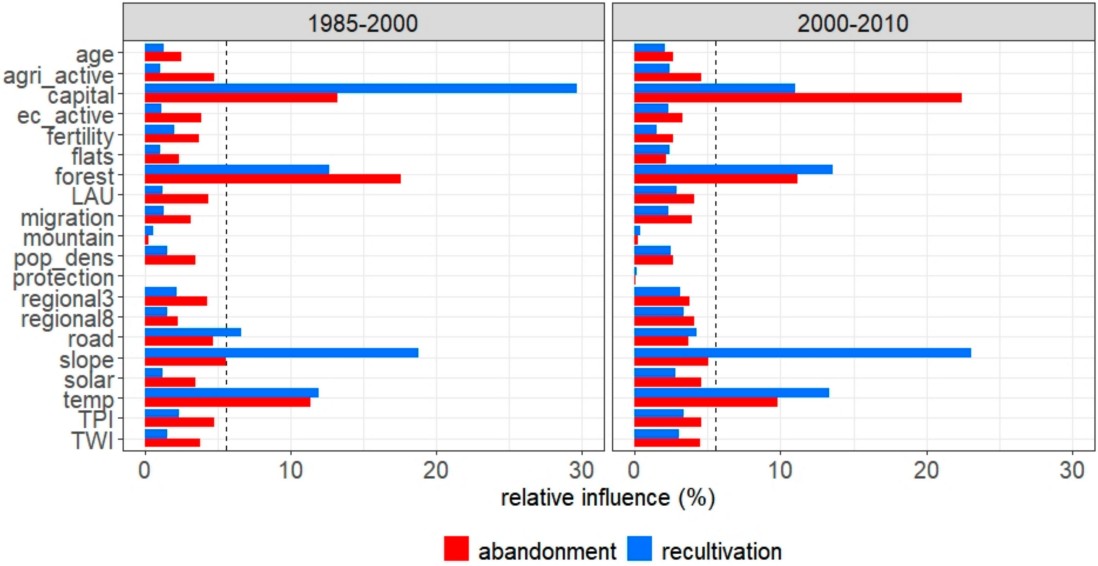

**Figure 5.** Relative influence of variables used to model agricultural abandonment, and recultivation. The dashed line indicates the threshold used to consider variables as meaningful for interpretation.

Recultivation in Slovakia represented primarily the conversion of grassland to cropland. Recultivation rates (proportion of areas from the total extent of agricultural land) were similar

in both periods (period #1, period #2) and comprised 2% of agricultural land (40,654 ha and 41,067 ha in the periods 1985–2000 and 2000–2010, respectively). Recultivation primarily occurred in the flatlands of eastern Slovakia, where abandonment was also common. In general, only ~6% of recultivated land in period #1 and ~11% period #2 occurred in mountainous areas (Figure 4).

Similarly to abandonment, recultivation was influenced by distance-based factors, temperature and slope steepness. However, the direction of the different factors' influence, which statistically explained recultivation patterns, was opposite to that of abandonment (e.g., recultivation increased with temperature, decreased with slope steepness and the proximity to forest patches), except for the distance to the capital city (Figures 5 and 6). A flatter slope accounted for more than 10% of the variance in recultivation in period #1 and 15% in period #2. The likelihood of recultivation was higher with increasing distance from the forest edge, with a higher average temperature and nearby roads. Similarly to abandonment, recultivation was more common further away from the capital city, but the relative contribution of the distance to the capital city dropped from 26% (period #1) to 8% (period #2). Recultivation in period #1 was also clustered in areas where recultivation was common in period #2. The AUC value was 0.94 for period #1 and 0.87 for period #2, respectively.

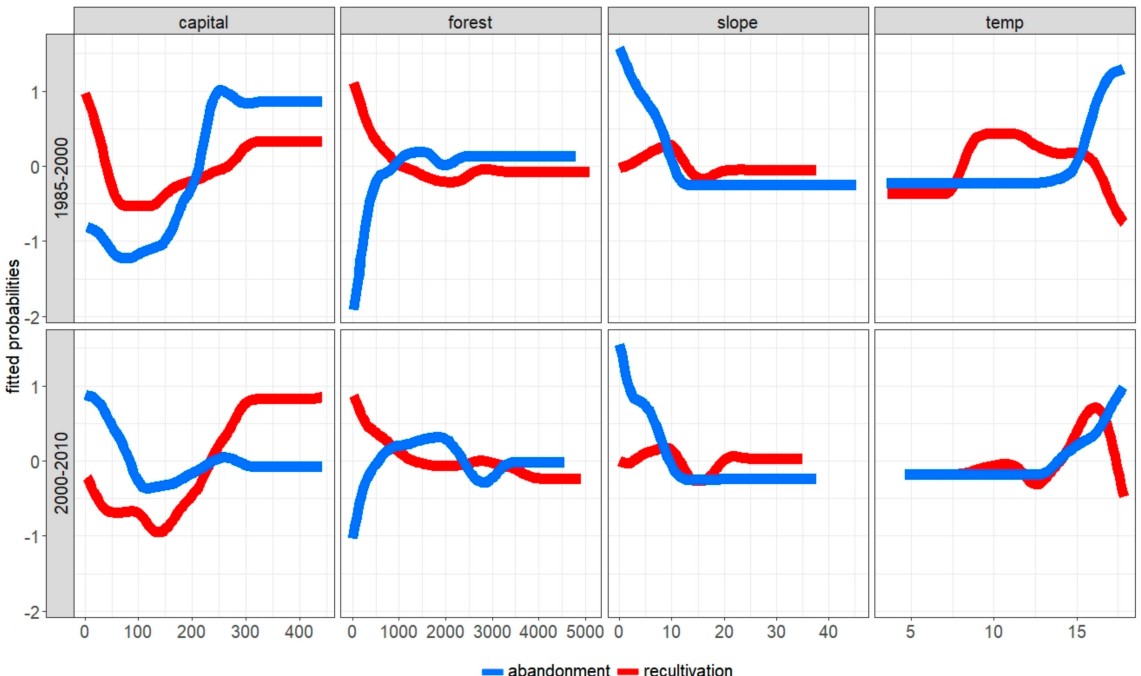

**Figure 6.** Fitted probabilities of abandonment of agricultural land (red line), and its recultivation (blue line) and their response to response to increases in distance to the capital city, distance to forests, slope and temperature. The x-axis values correspond to the measured ranges of particular variables.

### 3.2. Scenario of Future Agricultural Land Use Change

The long-term effects of agricultural land development were modeled by a climate-change scenario for 2060 using the same set of spatial determinants and extent of the abandoned area as was used to construct the model of agricultural land use change for period #2.

The scenario model outputs for 2060 suggest that both abandonment and recultivation, assuming a steady-state agricultural land use change decision-making process, are likely to occur particularly in eastern Slovakia, following an overall East-West gradient, and in proximity to forests, which also are more common in central and eastern Slovakia (Figure 7). Most suitable areas of agricultural land abandonment appeared in response to distance-based factors and climate change in the central mountainous parts of the country and flatlands in the eastern part of Slovakia.

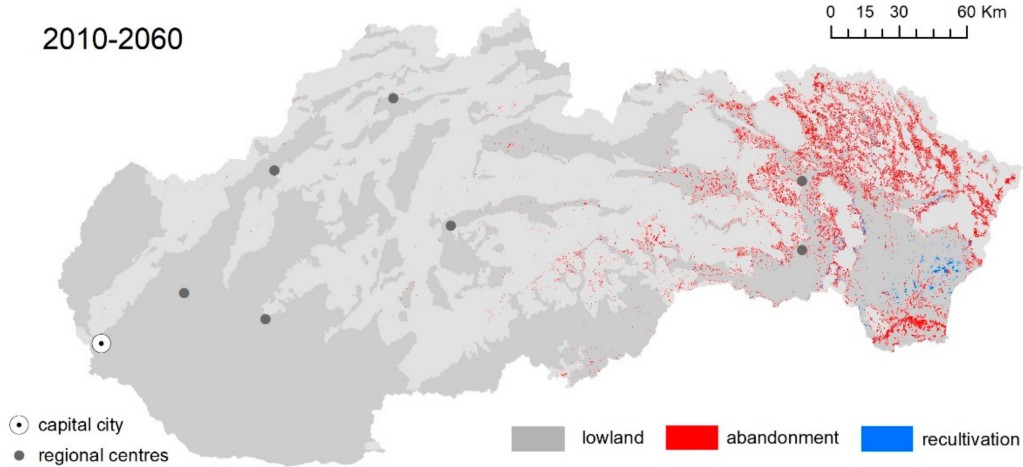

**Figure 7.** Future scenario of agricultural land use change in 2060.

## 4. Discussion

Our study has revealed widespread agricultural land abandonment in Slovakia, particularly in the plains that are suitable for farming. The share of abandoned agricultural land was twice as high in the mountains compared to the plains. Marginal mountainous areas were more prone to abandonment, which is in line with other studies [5,90,91].

### 4.1. Determinants of Agricultural Land Use Change

Proximity measures, such as the distance to the capital city and environmental factors, such as proximity to forest edges, increased slopes and lower temperature, determined the spatial patterns of agricultural land abandonment and recultivation across Slovakia. The non-parametric BRT models revealed interesting non-linear patterns in the relationships between determinants and agricultural land use change. For instance, the likelihood of agricultural land use change (both abandonment and recultivation) continuously decreased within a travel time of 1.5 hours from Bratislava, but increased again further away, leading to hotspots of abandonment and recultivation in eastern Slovakia. Agricultural land abandonment in the vicinity of Bratislava was likely related to land speculation [92] and ongoing urban sprawl of new housing areas [92,93]. Similar positive relationships between increased abandonment rates and proximity to important cities have also been found in Switzerland [76], Romania [94] and across the entire European Union [92,95]. In our case, the explanatory power of distance to regional centers as a variable explaining the agricultural land use change was higher than in other countries, such is in western Ukraine [43], Romania [75] and temperate European Russia [20]. This could be partly explained by the noncentric position of the capital city at the extreme south-western part of Slovakia and the country's longitudinal shape.

Distance to forest edges was a strong predictor of the spatial pattern of abandonment and recultivation in both study periods. Approximately 80% of abandonment took place within a distance of 1 km from forest edges. Remaining forests often occupy areas unsuitable for farming in Europe [40,96] and land in proximity to forests also tends to be less suitable for farming. Additionally, seed dispersal may also foster forest regrowth [76,97,98]. Land abandonment in proximity to forest edges can also be an intentional measure of forest defragmentation efforts that aim to optimize habitat connectivity and migration, especially near protected areas. However, we did not find evidence of such an effect linked to protected areas. Neighbourhood effects and protection of sites may also account for the opposite, positive role that increasing the distance to forests plays in agricultural recultivation. The relative influence of this variable is 13% for the transition period (period #1) and 14% for the EU-related period (period #2).

Steepness of slope did not play a major role for abandonment, but was important for explaining recultivation, which was more likely to be found at locations with gentle slopes (up to ~8°) (Figure 5). The explanatory power increased over time (19% for the transition period and 23% for the EU-related period), likely the result of the greater suitability of flat land for intensive and heavily mechanized agriculture. Such relationships between agricultural land change and terrain properties have been documented in previous studies in Slovakia [40,99,100], but also in other post-socialist countries [19,43], at the pan-European scale [5,79] and in global studies [101–103]. The reason for the low influence of terrain properties with respect to the abandonment patterns in our study likely relates to the dominant role of socioeconomic drivers, such as proximity factors [104].

The influence of temperature on agricultural land use change patterns is likely related to expected crop yields [68]. Locations with lower temperatures were more likely to be abandoned and less likely to be recultivated. Liberalization of prices after the transition to a market-oriented economy led to lower product prices and reduced the profitability of farming, particularly in less productive areas [105]. Therefore, areas with lower crop yields or less accessible areas (as expressed by the distance to roads) were more likely to be abandoned.

None of the demographic factors included in our models substantially influenced recultivation patterns. However, the demographic and socioeconomic conditions in Slovakia show a west-east gradient (e.g., we found a positive relationship between the proportion of economically active people and distance to the capital city (Pearson correlation coefficient r = 0.36)), and therefore the explanatory power of demographic variables could be masked out by the influence of factors such as distance to the capital city. The small explanatory power for demography factors is likely a sign that the analyzed factors are not important determinants for recultivation. The lack of a relationship between demographic factors and agricultural land use change has also been found in the review of other agricultural land change studies across Europe [62] and overall land change studies in Slovakia [58]. This may be due to the fact that private investments and CAP, which support the recultivation of marginal areas and private companies, act in a globalized way and do not consider demographics when investing in a particular region. Additionally, with the use of modern technology, the number of people employed in agriculture is constantly decreasing, and agriculture is no longer a workforce-intensive activity [23,106].

### 4.2. Agricultural Land Use Change Scenario

The scenario of future agricultural land use change was developed based on a similar set of variables and land change rates found in period #2. The spatial pattern exhibited a strong west-east trend (abandonment and recultivation in the eastern part of the country). Future changes in agricultural land use will likely be related to complex terrain conditions and complex land use configuration (e.g., proximity to forest edges as the area is characterized by a mosaic of agricultural areas and forests) and to areas with a relatively high increase in drought compared to other parts of Slovakia, as predicted by future climate models (RCP 8.5).

### 4.3. Accuracy of Land Change Assessment

For past agricultural land use change models, greater explanatory power (AUC score) was obtained for recultivation models. This implies that abandonment was a more complex process, with less of a connection to the natural and socioeconomic factors included in the model. This complexity may be due to more individualistic and local decisions on abandonment compared to recultivation, which are often driven by more global decision-makers. It is possible that the economic value of land and its attractiveness for further development (housing, industrial, infrastructure) are important local components, which may not be adequately considered in our models. Data at the farm or parcel level (e.g., amount of cultivated land; land titling) would provide additional insights into abandonment patterns. However, such data are not available and would require modeling approaches at different spatial scales compared to those used in this study.

Previous studies have shown that besides natural constraints, demography and location-based factors, abandonment is widely driven by ownership status and functioning land markets [28,47,107], field size [92,108] or land use legacies [109]. The lack of such detailed data at the national scale did not allow us to include these factors in our study.

Furthermore, distant connections and remote causes also play an important role in agricultural land use change [62,110]. For example, the global market demand for energy crop production changes the cropping structure (area used for different crops), as described in the introduction. Similarly, the recultivation pattern in certain areas could be influenced by demands on animal and feed production [111]. Considering the recultivation tendencies on the abandoned areas, farmers could choose the most suitable area from the entire pool of available land [96] following, eg., the physical properties of the landscape. This may have contributed to the high accuracy of our recultivation models and suggests a likely continuation of such a trend in the future.

### 4.4. Limitations of Our Models

The data we utilized for agricultural land use change has several potential limitations related to the scale and classification approach developed over the region spanning different countries in the Carpathian ecoregion according to [27]. False change detections may occur, for example, if large agricultural fields become fragmented over the study period. Such fragmentation could influence the reflectance captured by satellites and result in a mixture of different landscape components being falsely detected as some of the land changes used in our analysis (e.g., cropland to pastures [27]).

We considered several fine-scale spatial determinants in the statistical analysis of agricultural land use change. Although these determinants allow us to pinpoint the location of the changes we assessed, namely, abandonment and recultivation, they fail to provide direct insight into the strength and direction of underlying drivers for these changes, such as changes in agricultural subsidy schemes, land reforms or zoning measures. For these underlying drivers, data are not available in a spatially explicit form, such as the subsidies paid in small administrative areas. Our interpretation of the drivers of the observed changes, therefore, rests on our expert knowledge and the literature on land use changes in the region.

Before modeling, we removed highly mutually correlated spatial determinants from the list of variables (Pearson correlation coefficient above 0.7). Even after applying such filtering, the relative influence of the spatial determinants may still be masked by one another. Such interaction effects likely mutually decrease the explanatory power of both proximities of regional centers variables, or proximity to the regional centers vs. the proximity to district centers (LAU). These interactions, as one would expect, relates to the hierarchical levels that define the different categories of settlement. To a smaller extent, such interaction may be also found between the proximity to settlement measures and terrain properties as terrain also determined the calculated proximity measures.

For the scenario models, we extrapolated the agricultural land use change trends from the period 2000–2010. The allocation of the future agricultural land use change is determined by the factors influencing the agricultural land use in a period with persisting legacies of communism (e.g., ongoing restitution of agricultural land), or influenced by the accession to the European single market and CAP. This are likely events that will not appear again in the future. That means that the future pathways of the agricultural land use might be determined differently by the selected explanatory factors. However, trajectory from 2000 to 2010 better represents a mix of open-market economy and CAP subsidies and its likely that the market oriented tendencies will continue to drive the agricultural management in the future by, for example, selecting the most attractive areas for recultivation or abandonment of the marginal areas. Very different development trends and future pathways may occur if sudden, non-linear shifts in the land system were to occur [112]. These might be driven by an increase in demand for energy crops, different shifts in the climate system (different RCP scenarios) or abrupt changes in the national CAP implementation. For our trend scenario, we only considered the climate change scenario that corresponds to the upper boundary of greenhouse gas emission

trajectories (RCP 8.5), which leads to a temperature increase by 2100 of around 8.5 °C [113]. Our future scenario, therefore, should only be used for informative purposes, as it helps to better understand the implications of trends in land use change on agricultural land. From the regional and national land management perspective, those locations with a high risk of future agricultural land-cover change represent valuable high-natural value farmlands, and thus, additional efforts should be taken to adjust management strategies in those areas.

### 4.5. Implications of Agricultural Land Change

Agricultural land dynamics vary substantially within countries [20,114]), as well as between countries [8,13,115]. A nationwide assessment of past and future agricultural land-cover change, as presented in this study, reveals overall trends that may be hidden in studies conducted at the local or case study level [13,116].

At the national level, future trends underpin the prevailing polarization of rural agricultural land-cover, which may be the result of the recultivation of lowlands and abandonment of farmland in the mountains [62,117,118]. The effect of climate change, however, may shift recultivation towards the mountains. The polarization effect showed that the agricultural support schemes applied within the period 2000–2010 only had a limited effect. This limited effect of support schemes has also been found in other European mountainous regions [119,120]. Current support schemes for less-favored areas in Slovakia take altitude, slope, average yield, population density, the proportion of people working in agriculture and soil properties into consideration [121]. Although we did not consider the relationship between yield and abandonment, agricultural land-cover changes correlated well with biophysical factors. For example, we found that slope steepness was strongly related to abandonment and recultivation patterns. Our results also reveal that the proportion of people working in agriculture accounted for less than 5% of abandonment patterns. The decoupling of population and farming activity could become even stronger in the future, as future technological improvements and digitalization of farming will lead to fewer people being involved in farming. In contrast, the distance to the capital and detailed climatic characteristics did account for agricultural land-cover change patterns. Currently, these factors are not taken into account in EU support schemes. We propose that including such spatially detailed environmental and proximity indicators (the identification of marginal economically marginal regions) would improve the effectiveness of such support schemes considerably.

## 5. Conclusions

We analyzed the spatial patterns and determinants of agricultural land change for all of Slovakia during the transition from socialism and EU accession periods and predicted agricultural land-cover change patterns until 2060. Abandonment dominated during the transition period (1985–2000), occurring on 10.9% of all agricultural land as a consequence of the economic shock caused by open markets and the decline in agricultural subsidies. The effect of EU support on agriculture was reflected in the second period (2000–2010) when agricultural land abandonment rates decreased to 5%. Recultivation remained at the same level in both periods (2% of agricultural land) and was concentrated in locations with similar agro-environmental characteristics defined by a relatively warm climate and flat terrain.

The selected natural factors and distance-based factors were found to predict recultivation more accurately than the abandonment of agricultural land-cover. Lower accuracies for abandonment, together with the higher dependency of this process on distance-based factors than natural conditions, indicate that abandonment of agricultural land-cover in Slovakia was a more complex process than recultivation of agricultural land, which largely depended on climate and terrain conditions.

The effect of removing the variance unexplained by our models (inaccuracies of the models compared to the observed patterns) could be seen in a plausible future scenario of agricultural land-cover change by 2060 predicted by models based on rates of agricultural land-cover change measured in the period of 2000–2010. Under this future scenario, abandonment and recultivation clusters in less

accessible mountainous areas, while recultivation also takes place in areas with comparatively better terrain and climatic conditions. Assuming that temperatures increase in the future, areas suitable for recultivation would be found in semi-natural mountainous areas which were abandoned and are currently not considered to be attractive for agricultural production. Giving more weight to the accessibility of the agricultural areas at different regional administration scales (i.e., district-level, country-level) in governmental support schemes may help to bring such areas back to cultivation and prevent the loss of the cultural value of such landscapes, or loss of agricultural areas that are particularly rich in biodiversity, such as semi-natural grasslands.

**Supplementary Materials:** The following are available online at http://www.mdpi.com/2073-445X/9/9/316/s1.

**Author Contributions:** Conceptualization, R.P., J.L., A.V.P. and D.M.; methodology, R.P., D.M., A.V.P. and Z.Z.; formal analysis, R.P.; investigation, R.P., J.L.; data curation, R.P., J.L., Z.Z. and T.L.; writing—original draft preparation, R.P., J.L., A.V.P. and D.M.; writing—review and editing, R.P., J.L., A.V.P., D.M. and M.B.; visualization, R.P. All authors have read and agreed to the published version of the manuscript.

**Funding:** This research was funded by EU FP7 ERA.Net Russia Plus (grant number: 449 CLIMASTEPPE), DFF-Danish ERC Support Program (grant number: 116491, 9127-00001B), the Slovak Scientific Grant Agency VEGA under Grant No. 2/0023/19 "Land cover dynamics as an indicator of changes in the landscape" and the Slovak Research and Development Agency, project No. APVV-17-0377 Assessment of recent changes and trends in agricultural landscape of Slovakia.

**Acknowledgments:** We thank Patrick Griffiths for the provision of the agricultural land-cover change dataset, Tobias Kuemmerle for providing comments and suggestions on the early version of the manuscript and Curtis Gautschi for professional language editing and four anonymous reviewers for their useful comments.

**Conflicts of Interest:** The authors declare no conflict of interest. The funders had no role in the design of the study; in the collection, analyses, or interpretation of data; in the writing of the manuscript, or in the decision to publish the results.

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
