# Peer review of "Abandonment and Recultivation of Agricultural Lands in Slovakia—Patterns and Determinants from the Past to the Future"

_land, doi:10.3390/land9090316_

Round 1

Reviewer 1 Report

The manuscript “Abandonment and recultivation of agriculture in Slovakia – patterns and determinants from the past to the future” analyzed the patterns and determinants of agricultural land abandonment and recultivation in Slovakia during the transition from a state-controlled to an open-market economy (1986 to 2000) and the subsequent accession to the European Union (2000 to 2010). The manuscript used 30-m multi-seasonal Landsat imagery to quantify agricultural land-cover change and analyzed the socio-economic and biophysical determinants of the observed changes using boosted regression trees. In addition to analyzing past trends and determinants, the manuscript used a scenario-based approach to assess future agricultural land abandonment and recultivation. By doing so, the manuscript provided holistic and integrated insights on agricultural land use dynamics both in the past and possible scenarios in the future. The manuscript is well written, technically sound and will be very relevant to the readership of the journal and I would recommend its publication after issues highlighted below are adequately addressed.

My major concern with the manuscript is that after a closer look and assessment, one can realize that some contents or parts of the manuscript are not original as it is presented in the abstract. The results or findings for the stated first objective (using 30-m multi-seasonal Landsat imagery to quantify agricultural land-cover change) were taken from a published peer review articles (see references 27). In addition, information in section 2.2 and section 3.1 (Line 308-320) are from reference 27.

The originality of the manuscript is the analysis determinants of past land use changes using Boosted regression trees and modeling future land use change scenarios.

I would recommend that the authors revise the study objective by explicitly focusing on analyzing determinants of agricultural land use change and future scenarios. For that, they have used land use cover change analysis from a previously published paper or they build on previous work. Currently, it is not clear particularly in the abstract. Consequently, the last sentence of the abstract needs to be revised as the manuscript did not analyze directly earth observation data.

My second concern is about how the authors “match” the resolution of the different data used. Landsat imagery has 30m resolution while other data used (DEM, topographic wetness index, soil map, climate data) are in different resolution. Currently, it is not clear in the section 2.3. Please clarify.

Figure 6: Please increase the width of the legend line (red and blue). It is not easy to identify which one is red and which one is blue.

I recommend a revision of the title: I would suggest the terms “recolonization” instead of “recultivation of agriculture”. The title would then read: “Agricultural land abandonment and recolonization in Slovakia – patterns and determinants from the past to the future

Line 36 (Abstract): edit out “compared”

Reviewer 2 Report

This is an interesting and well-written article concerning historical land changes that are fairly well-contextualized. The scenario analysis or projection of these land-change trends, however, lacks some justification. While the authors provide necessary caveats about extrapolating land-change trends based on past observations, they neglect to specify the details about how the projection of trends is performed.

The technical approach seems to be statistically sound and uses appropriate model validation techniques. However, essential details on the scenario modeling are missing. In particular, the "allocation model" lacks any real description that would enable reproduction of the methods. There are (potential) substantive or conceptual issues with the allocation model, too. Demand for agricultural land use is perhaps the most important factor driving future land-cover/ land-use change (LCLUC), but it's not clear that demand is accounted for at all. The approach to allocation is described as in "the steady state" but there are multiple ways that could be operationalized and any one of them requires justification, which is missing.

Ultimately, if the authors are sufficiently modest about the extrapolation of land-change trends, the current results are probably fine and do not need to be revisited. However, critical detail on the scenario modeling, particularly related to allocation, is required.

Lines 87-106: The background information on land-cover/ land-use change (LCLUC) mapping attempts in Slovakia or greater Europe is a stumbling block for readers. I suggest moving much of this information to the "Data and Methods" section. This will allow readers to more quickly develop an understanding of the importance and objectives of your study while reading through the Introduction.

Lines 107-112: Similar to my previous comment, this paragraph contains technical details that probably belong in the "Data and Methods" section.

Lines 116-117: You write: "Quantification of these drivers then
permits land-cover change to be forecasted." However, you previously identified accession to the EU as a "trigger" for agricultural land-cover change in CEE countries (Lines 69-70). The collapse of Soviet socialism is implied as a driver of agricultural land "transformation" (Lines 59-61). These events will not be repeated and similar events might not occur within the time frame of your forecast. I think you need to add a caveat. While the various data sources on land conditions (e.g., socio-economic factors, land ownership, proximity, etc.) that potentially drive land change (which you are about to list in the next section) are probably thoughtful and chosen well, the overarching socio-political drivers are unique. Your forecast model, having been trained on unique observations (post-Soviet socialism transformation and accession to the EU), may not be very predictive of the future. This caveat may not belong to the Introduction (rather, in the Discussion) but it is something I want to bring up early as you develop this narrative.

Line 149 and Figure 1: Because this is a land-cover change study, it might be more effective to have a thematic map that shows Slovakia in detail. You describe how "the landscape can be divided into two geomorphological
parts, which also reflects the specialization of agriculture" (Lines 136-137) and how "a concentration of agricultural production areas are found in the Danubian Lowland, located in the south-west of Slovakia" (Lines 140-141), so it might be better to show these details to readers using a large-scale map of Slovakia instead of whole Europe. The location of Slovakia within Europe could be shown as an inset panel.

Lines 156-158: I think you could rewrite this sentence (beginning "At the same time...") to make it more clear that the very small amount of non-cropland conversion to cropland is what serves as "a sign of the intensification of agricultural production." Or adding the qualifier "only 0.4%" might make this clear.

Lines 176-180: It's unclear whether these are maps of land cover (current land-cover conditions) or land-cover change. It's also unclear whether the land-cover categories are broad or specific to agriculture. The paragraph begins: "We utilized 30-m land cover maps..." but includes the phrase "the maps of agricultural land-cover change." Please re-write this paragraph to make it clear the land-cover categories in the classification and the dates used to define land-cover change (i.e., start date and end date).

Lines 184-186: You write: "Even though the conversion of croplands to grasslands is a certain first sign of reduced land-use intensity, we assumed such a transformation to be a precursor of abandonment." It's not immediately clear what the distinction is. Perhaps you could re-write this sentence to make it clear that an alternative interpretation of cropland-to-grassland conversion is reduced intensification (e.g., letting a field lie fallow).

Lines 176-188: On first read, it seems like this is all you intend to say about the land-cover classification. But I now think you intended to "sign-post" or to provide an overview of your analysis--which is very helpful to the reader, but only if it is put in the right place. Here, it comes right after "Study Area" so it seems like it is a detailed description of a step in your analysis, rather than an overview. I suggest moving these lines up so they are the body text of the "Data and Methods" and adding language, e.g., "Our analysis consists of..." or "For this study, we conducted the following analyses: ..." And then, this section could provide more (necessary) detail on the land-cover classification. Otherwise, one gets the impression that you used some land-cover (change?) classification from a previous study, which is why you provide so little detail.

Lines 195-196: You write: "The DEM was interpolated to the 15-m grid..." but it's not clear what "the 15-m grid" is. Landsat data (which were mentioned earlier) are nominally available on a 30-m grid, unless pan-sharpened data is used.

Lines 198-200: Are these von Neumann neighborhoods (4-connectivity, or pixels sharing a side) or Moore neighborhoods (8-connectivity, or pixels sharing a vertex)? And I'm assuming "direct" here means one of these two options. Please clarify in the text.

Lines 203-205: As reference [74] doesn't appear to be available to non-Slovak readers, please provide some more details on how soil fertility was calculated.

Lines 250-255: The grammatical issue aside (see below), this short literature review probably isn't really necessary for a technique like boosted regression trees, which is fairly well-established in land-change science.

Line 304 and Table 1: Some entries for the accessibility-related determinants read: "path distance in minutes the friction of LU classes and DEM." I'm not sure what this means. I would think the "path distance in minutes" and "the friction" are two different things, but perhaps you mean that the "path distance" is used as the "friction" (I think the word "cost" is more commonplace in routing applications, as in "least-cost path")? Please clarify this in Table 1 and also on Lines 218-226. It's a little hard to figure out exactly how you calculated accessibility.

Lines 256-261: Please describe the software you used to fit the boosted regression trees.

Line 266: "A similar number" or the same number? It should probably be the same (i.e., equal) number of samples.

Line 272-273: Please make it clear that "interaction levels" correspond to the depth of the tree/ number of branches or splits. I think that ensemble decision trees are actually fairly intuitive, but the language you are using here makes it harder to understand. I would suggest using tree-related terms: nodes, branches, etc. I know you cited Hastie et al. (2009); I think the more approachable reference James et al. (2013) by some of the same authors uses more intuitive language to describe boosted regression trees [James, G., D. Witten, R. J. Tibshirani, and T. J. Hastie. 2013. An Introduction to Statistical Learning with Applications in R. New York, New York, USA: Springer Texts in Statistics.].

Lines 283-288: How are the "probability map[s]" generated? I assume they come from the boosted regression tree model, but the reader would benefit from some specificity here (or in Section 2.5). Also, I'm assuming that the probability maps are used to draw a prediction of the land-cover change for each pixel at each time step. Please be more specific about this *simulation* approach; "we iterated" is a little too generic of a description. Finally, the "simple allocation model" needs more detail. While it may be conceptually simple (though I am forced to make assumptions here) because it is somehow related to demand for agricultural land uses(?) the execution of this allocation model is a very important detail. On Lines 288-289 you write "the allocation process were considered to be static;" does this mean that in each iteration the model requires that the total area of abandoned land is balanced by an equal area of re-cultivated land? Much is unclear about this process.

Line 320 and Figure 3: Since the darker gray color corresponds to "arable land," I'm assuming that the lighter gray color corresponds to non-arable land. Please make this clear in the Figure 3 caption or in the map legend.

Lines 366-368: You write: "The scenario model outputs for 2060 suggest that both abandonment and recultivation are likely to occur particularly in eastern Slovakia..." It's hard to evaluate the accuracy of this statement since it is unclear how demand for agricultural land through 2060 is determined. For example, if the population of Slovakia increased considerably through 2060 and food requirements were not met by imports, there might be no agricultural land abandonment because more land is needed to grow crops. Until those details are available, it seems more accurate to say that these areas (Figure 7) are the areas most *suitable* for abandonment or recultivation, assuming a steady-state abandonment/recultivation decision-making process.

Figures 3 and 7: It would help the reader understand the salience of your Discussion if these maps had city and/or provinces clearly identified. Without city/ province labels, it's hard to match your interpretation of drivers (e.g., "Agricultural abandonment in the vicinity of Bratislava was likely related to land speculation...and ongoing urban sprawl of new housing areas," Lines 386-387) with observed spatial patterns.

**Minor typographical/ grammatical edits:**

Line 65: For clarity, I would suggest revising to: "differences in the speed and depth of the implementation of reforms following the demise of socialism"

Lines 66-67: Better to start this sentence ("Abandonment tended to be higher...") with "For instance" instead of putting it in the middle of the sentence: "For instance, abandonment tended to be higher..."

Line 129: "quantify the influence of determinants on abandonment and..." might be more clear written as: "quantify the determinants of abandonment and..."

Line 190: "farmers' decision" should probably by "farmers' decisions" (plural)

Line 246: "Similarly to" might read better as: "As with" or "In common with"

Lines 250-255: This sentence needs to be revised for parallel construction. It might be easiest to put a colon after "BRTs have been used to:" and then change each subsequent clause so they can be paired with this statement, possibly separated by semicolons. In particular, the first ("determine the factors...") and third ("identify the land use factors...") work well with "have been used to" but the second clause ("those associated with disease transmission...") does not work (it would read: "BRTs have been used to...those associated with disease transmission").

Line 283: "the prediction allowed to address" should probably be "the prediction allowed us to address"; but would be more clear still written as, e.g., "prediction of climate-change effects is possible."

Line 392: "excentric" should probably be spelled "eccentric" but I don't think this is the right word to use... Perhaps the authors meant "exceptional"?

Lines 425-428: You note there is a correlation between accessibility and demographic factors. Random forest models are impacted by multi-collinearity in the importance metrics (though accuracy of predictions is not impacted). I assume the same is true for boosted regression tree models. Did you make any attempt to test the impact of multi-collinearity on the variable importance? For instance, removing one of a pair of correlated variables and re-fitting the model?

Lines 458-459: "changes the crop structure..." What is crop structure? Do you mean the relative cropping areas of different types of crops?

Lines 459-460: What is "the globalization of recultivation tendencies"? Are you saying that re-cultivation is a common trend across countries? And yet Slovakia has shown much more abandonment than re-cultivation in the past. This needs some explanation and justification.

Reviewer 3 Report

Altogether, I think this is an interesting article with an innovative methodology. There is some scope for improvement:

Keywords

  • „Drivers“ should be changed to a more specific keyword, e.g. land-use drivers.

  • Land use does not need a „-“ (except it is combined with further words). The same is true for the rest of the text. Besides, in line 49 it has a „-“, in line 50 it does not.

1. Introduction

  • Maybe „land abandonment“ should be defined as it can be understood in different ways.

  • Reference 1 is from 2005. I would recommend to add some recent, agricultural-related references. I also recommend to add climate change in sentence 2.

  • line 52: Agriculture is not only responsible for GHG in tropical regions. As the text is not about tropical regions, this sentence is unnecessary.

  • The terms land-use change and land-cover change are alternated without explanation→ definitions and harmonization are needed.

2. Data and Methods

  • 2.2 It seems daring to me to see the transformation of arable land into grassland as a precursor to land abandonment. Do the authors have specific reasons to assume this, is there respective data to rely on? This is particularly questionable because the other way round it is assumed that the conversion of forests to grassland is understood as recultivation. This defining issue of course has a strong impact on the results.

  • 2.3 Since [74] is a slovak reference, it’s hard to verify if the understanding of soil fertility as a function of particle size distribution and soil type is sufficient. It seems vague to me. What’s about soil nutrient content, historical and recent management practices etc.?

  • Line 212: What are protected areas? Since authors mention the Ramsar Convention, they also should mention the European Natura-2000-Directives.

  • Line 218: It’s not clear why travel costs are relevant → A critical reflection of the chosen parameters/ an explanation why parameters are chosen would be fine – maybe before the discussion chapter, because it’s an important part of the methodology. How are the parameters identified, e.g. by literature review?

  • 2.4. Language check is particularly necessary for this subsection.

  • 2.5 The proportion of errors between mapped and predicted areas would be interesting.

3. Results

  • It seems surprising to me that soil fertility (of cause depending on its definition) is less important for recultivation then e.g. distance to the capital or temperature. And why only Bratislava is considered in the analyses but no other large cities?

  • Line 428: The authors say hat other studies across Europe came to similar results but there is only one single reference.

  • Energy crop production may not only change the crop structure but also trigger land-use changes. The authors may give some more information on this issue. Besides, rising consumption of animal-derived products may also lead to land-use changes due to rising feed production.

  • Line 482 → “national CAP policy” → The Common Agricultural Policy is an European policy, Member States have some scope for implementation. Maybe “national CAP implementation” would be better.

  • Why did the authors choose the RCP 8,5 – an temperature increase by 2100 of around 8,5 °C would have strong implications on agriculture, which are not reflected by the authors in their future scenario. By the way, I think the article would be interesting even without the future scenario.

  • How should spatially detailed environmental and proximity indicators be implemented in the CAP subsidy scheme? It’s a vague recommendation at the end of section 4 and I’m not convinced if this would really improve the effectiveness of the CAP. E.g., should the distance to be capital be a factor that increases or decreases the subsidies and why this would be positive?

5. Conclusion

  • Here, some more information on the “governmental” support of certain areas is given. That’s an interesting point. But authors recommend to give subsidies to agricultural areas with particular biodiversity value. This is in contrast to the fact that agriculture is one of the main drivers of biodiversity loss (what the authors indicate at the beginning of the article), even if this depends on agricultural practices.

Reviewer 4 Report

This study takes the moves by the consciousness that changes in agricultural land use have been the main cause of global environmental change, land degradation, biodiversity losses and the reduction of ecosystem services. For this reason, the study offers a method to better understand the spatial consequences of agricultural land-cover change.

The paper is well written and well structured, clear in the exposition and in the definition of the objectives. The obtained results are interesting, and, in my opinion, the paper can be accepted.

The study is well thought, supported by a good scientific literature. I’d appreciate an examination of the theories that, in some way, already treated the proposed idea; it would be better to create a paragraph about the state of art.

I think that the research is on the right way and that it will offer food for thought for future research in this field.

Round 2

Reviewer 1 Report

I have carefully read the revised manuscript and authors’ responses. I am happy with most of the improvement and the answers. The manuscript has been adequately revised and warrant a publication.

The authors respond to my recommendation on the term “recultivation of agriculture” must be improved. The terms “Agricultural colonization” or “Agricultural recolonization” are well known concepts in agrarian studies. The term “Recultivation of agricultural lands” might be more appropriate than “recultivation of agriculture”, which is less meaningful. I suggest:

“Agricultural lands abandonment and recultivation in Slovakia – patterns and determinants from the past to the future”

OR

“Abandonment and recultivation of agricultural lands in Slovakia – patterns and determinants from the past to the future”

Previous papers on the subject in the region are in line with my recommendation. See references below

  • Smaliychuk et al 2016 Recultivation of abandoned agricultural lands in Ukraine: Patterns and drivers. Global Environmental Change 38:70-81.
  • Estel et al 2015 Mapping farmland abandonment and recultivation across Europe using MODIS NDVI time series. Remote Sensing of Environment 163:312-325
  • Meyfroidt et al 2016 Drivers, constraints and trade-offs associated with recultivating abandoned cropland in Russia, Ukraine and Kazakhstan. Global Environmental Change 37:1-15
  • Zupanc et al 2016 Recultivation of Agricultural Land Impaired by Construction of a Hydropower Plant on the Sava River, Slovenia. Land Degradation & Development 27 (2):406-415

Reviewer 2 Report

The authors have added important details about their methodological approach and, in particular, clarified the land allocation approach used to quantify future abandonment and recultivation of agricultural lands. Some of the details in the Response to Reviewers should be added to the manuscript itself.

I am particularly concerned about the sensitivity of the variable importance metrics to multi-collinearity. The authors responded to this topic in the Response to Reviewers but did not revise the manuscript. They wrote (in the Response): "Prior to running the BRTmodel, we tested the selected variables on multicollinearity. The pairwise comparison of variables showed, there were no pairs of variables with high correlation (we set the threshold for R2 to 0.7)." This is a high threshold for correlation; pairwise correlations lower than this threshold could still mask one another in the regression tree ensemble. While I think it would be prudent to test for this formally, it would be sufficient if the authors addressed this concern in the section on Limitations. Specifically, I would like to see a brief discussion of which variables are highly correlated with one another and how that might impact the variable importance metrics (Figure 5), i.e., are there any variables that potentially have a lower relative influence because their effect is being masked by another?
